# Combining ReACp53 with Carboplatin to Target High-Grade Serous Ovarian Cancers

**DOI:** 10.3390/cancers13235908

**Published:** 2021-11-24

**Authors:** Adam Neal, Tiffany Lai, Tanya Singh, Neela Rahseparian, Tristan Grogan, David Elashoff, Peter Scott, Matteo Pellegrini, Sanaz Memarzadeh

**Affiliations:** 1Department of Obstetrics and Gynecology, David Geffen School of Medicine, University of California Los Angeles, Los Angeles, CA 90095, USA; neal.adam37@gmail.com (A.N.); tlai@mednet.ucla.edu (T.L.); tanyasingh@mednet.ucla.edu (T.S.); neelarahseparian@gmail.com (N.R.); 2UCLA Eli and Edythe Broad Center of Regenerative Medicine and Stem Cell Research, University of California Los Angeles, Los Angeles, CA 90095, USA; matteop@mcdb.ucla.edu; 3Department of Medicine Statistics Core, David Geffen School of Medicine, University of California Los Angeles, Los Angeles, CA 90095, USA; TGrogan@mednet.ucla.edu (T.G.); DElashoff@mednet.ucla.edu (D.E.); 4Department of Life, Earth, and Environmental Sciences, West Texas A&M University, Canyon, TX 79016, USA; pscott@wtamu.edu; 5Department of Molecular, Cell and Developmental Biology, University of California Los Angeles, Los Angeles, CA 90095, USA; 6Institute for Quantitative and Computational Biology, University of California Los Angeles, Los Angeles, CA 90095, USA; 7Molecular Biology Institute, University of California Los Angeles, Los Angeles, CA 90095, USA; 8UCLA Jonsson Comprehensive Cancer Center, University of California Los Angeles, Los Angeles, CA 90095, USA; 9The VA Greater Los Angeles Healthcare System, Los Angeles, CA 90073, USA

**Keywords:** ovarian cancer, p53, carboplatin resistance, combination therapy, in vitro organoid drug assay

## Abstract

**Simple Summary:**

Clinical management of ovarian cancer remains a major clinical challenge as many patients develop resistance to standard platinum-based chemotherapy drugs over time. Testing novel targeted strategies and combination therapies may open the door to new possibilities for the treatment of this disease. One such approach includes targeting p53 with a peptide called ReACp53. While mutations in p53 are common in many cancers, ovarian cancers, in particular, are characterized by the dysfunction of this protein. The aim of this study is to evaluate the potential of combining ReACp53 with standard platinum-based chemotherapy to target ovarian cancer tumor cells. Using in vitro and in vivo preclinical models, we demonstrate enhanced efficacy when combining ReACp53 and carboplatin to target a subset of ovarian cancer cell lines and primary patient tumor samples. Collectively, our results indicate that this combinatorial approach may be applicable for targeting human ovarian tumors.

**Abstract:**

Ovarian malignancies are a leading cause of cancer-related death for US women. High-grade serous ovarian carcinomas (HGSOCs), the most common ovarian cancer subtype, are aggressive tumors with poor outcomes. Mutations in TP53 are common in HGSOCs, with a subset resulting in p53 aggregation and misregulation. ReACp53 is a peptide designed to inhibit mutant p53 aggregation and has been shown efficacious in targeting cancer cells in vitro and in vivo. As p53 regulates apoptosis, combining ReACp53 with carboplatin represents a logical therapeutic strategy. The efficacy of this combinatorial approach was tested in eight ovarian cancer cell lines and 10 patient HGSOC samples using an in vitro organoid drug assay, with the SynergyFinder tool utilized for calculating drug interactions. Results demonstrate that the addition of ReACp53 to carboplatin enhanced tumor cell targeting in the majority of samples tested, with synergistic effects measured in 2 samples, additivity measured in 14 samples, and antagonism measured in 1 sample. This combination was found to be synergistic in OVCAR3 ovarian cancer cells in vitro through enhanced apoptosis, and survival of mice bearing OVCAR3 intraperitoneal xenografts was extended when treated with the addition of ReACp53 to carboplatin versus carboplatin alone. Results suggest that carboplatin and ReACp53 may be a potential strategy in targeting a subset of HGSOCs.

## 1. Introduction

Ovarian cancer is a deadly gynecologic malignancy responsible for over 13,000 deaths annually in the US [1]. Among ovarian cancer diagnoses, the most common subtype is high-grade serous ovarian carcinomas (HGSOC) [2]. Standard therapy for HGSOC includes surgical resection of the tumor followed by platinum-based chemotherapy with carboplatin [3]. Patients with advanced-stage disease not amenable to frontline cytoreduction are often first treated with chemotherapy (neoadjuvant chemotherapy) to reduce overall tumor volume, followed by an interval debulking surgery [4]. While many patients achieve remission with initial treatment, approximately 80% of patients with advanced-stage disease experience tumor relapse [3] which is associated with the development of platinum-resistant disease. Ultimately, it is platinum resistance that claims the lives of women diagnosed with HGSOC. Effective targeting of these tumor cells and prevention of disease relapse remain a clinical challenge.

While alterations in p53 are found in about half of all human tumors [5], mutations in this protein are especially prevalent in HGSOCs, observed in >90% of cases [6]. According to a recent study mapping the evolutionary history of many cancers, mutations in p53 may be a very early event in HGSOC development and can be detected many years before diagnosis [7]. Further, driver mutations in p53 may be an initiating event in HGSOC [8,9,10], and high levels of p53 immunostaining serve as a biomarker of early-stage disease in serous tubal intraepithelial carcinoma lesions. p53 plays an important role in maintaining the stability of the cell’s genetic information, guarding against genomic chaos [11]. Wildtype function of p53 is crucial as a tumor suppressor by acting as a cellular stress sensor, inducing cell-cycle arrest, and promoting DNA repair upon cellular injury or genotoxic damage. If a cell accumulates DNA damage too severe for repair, p53 activates an apoptotic program, which leads to the elimination of the damaged cells [5,12,13]. This function prevents genomically unstable cells from replicating and becoming cancerous. Because mutations in p53 are associated with chemotherapy resistance and potentially worse clinical outcomes [12,13], therapeutic targeting of this protein is a major goal of many researchers globally.

The clinical challenge of disease recurrence despite the administration of platinum-based chemotherapy has necessitated the exploration of alternative options for the treatment of ovarian cancers. Primarily, these alternatives include the addition of targeted agents to standard chemotherapy. Because p53 is a regulator of cellular apoptosis, restoration of p53 function may enhance platinum-mediated apoptosis. Such approaches are currently being explored clinically. For example, APR-246 has been tested in clinical trials in combination with cytotoxic drugs, such as carboplatin and doxorubicin (NCT:02098343), in treating patients with recurrent HGSOC. Further, the development of agents that target p53 is a growing field in cancer therapeutics. One such peptide, called ReACp53, has been shown to target multiple cancer models in vitro and in vivo [14,15]. While the exact mechanism of ReACp53 induced cytotoxicity remains unknown [16,17], it was designed to inhibit mutant p53 aggregation restoring its wildtype function (reviewed in [18]). The aim of the present study is to test the efficacy of ReACp53 in combination with carboplatin chemotherapy in targeting human ovarian cancers.

## 2. Materials and Methods

### 2.1. Cell Lines and Primary Patient Samples

Human ovarian cancer cell lines were obtained from ATCC or the National Institutes of Health (NIH), and frequently STR verified during experiments. All cell lines were maintained in recommended media (RPMI/10% FBS or DMEM/10% FBS) at 5% CO_2_ and 37 °C. This study was approved by the UCLA Office of the Human Research Protection Program and (IRB# 20-001762, IRB# 10-000727) and the VA Greater Los Angeles Institutional Review Board (IRB# 2019-020090). Clinical information from consenting patients was obtained from medical records. This heterogeneous population of patients consisted of tumors from patients who were chemo naïve, neoadjuvant chemotherapy-treated (both platinum-sensitive and platinum-resistant), and recurrent platinum-resistant. Platinum resistance was defined as tumor relapse <6 months from the last infusion of platinum-based chemotherapy. Platinum sensitivity was defined as tumor relapse >6 months after the final administration of platinum drugs. Solid tumor samples from consenting patients were obtained fresh from the operating room, brought back to the lab, and dissociated mechanically and enzymatically (collagenase 1 mg/mL and dispase 1 mg/mL, Gibco). Effusion samples were harvested by centrifugation and filtered through a 100 μM filter. All patient tumor samples were cryopreserved in FBS/10% DMSO in multiple aliquots to facilitate experimental repeats. 

### 2.2. Drug Preparation

ReACp53 peptide (amino acid sequence: RRRRRRRRRRPILTRITLE) used for the experiments outlined was either purchased from the Chinese Peptide Company or generously provided by Dr. Alice Soragni (UCLA). In either case, lyophilized ReACp53 was reconstituted in PBS (pH 8.5) at 5 mM and then sterile filtered for final use. 5-Carboxyfluorescein tagged peptide (5-FAM-ReACp53) was used to confirm cellular penetrance (Appendix A). The specificity of the peptide was tested in vitro using three cell lines (OVCAR3, SKOV3, and MCF7) using a previously reported high throughput in vitro organoid drug assay (Appendix A) [19,20]. Carboplatin (Tocris) was reconstituted in ddH2O at a concentration of 10 mM and sterile filtered for final use.

### 2.3. High Throughput In Vitro 3D Organoid Drug Assay

The efficacy of the ReACp53 and carboplatin combination was tested using a previously reported high throughput in vitro organoid drug assay [19,20] using human ovarian cancer cell lines and primary patient HGSOC tumor cells. In this assay, 5000 cells per well were suspended in a mixture of Matrigel matrix (Corning, Corning, NY, USA) and MammoCult medium (STEMCELL Technologies, Vancouver, BC, Canada) and plated around the rim of the wells of a 96-well plate. Following organoid establishment for two days, cells were treated with drugs for three days, replenished daily. Dose combinations were administered in triplicate wells for each plate (5 independent plates per cell line and 2 independent plates for each primary patient tumor sample). Cell viability was assessed using an ATP luminescence assay (CellTiter-Glo 3D Viability Assay, Promega, Madison, WI, USA) [19,20]. Viability percentages were calculated by normalizing each luminescence value to control wells (0 µM carboplatin, 0 µM ReACp53) using GraphPad Prism 8. Each cell line or primary tumor sample was plated by two independent investigators.

### 2.4. Measurement of Apoptosis Markers in Response to ReACp53 and Carboplatin Combination

To assess apoptosis in ovarian cancer cells treated with ReACp53 and carboplatin, 40,000 OVCAR3 cells per well were plated in 24-well tissue cultures dishes embedded in Matrigel (Corning) to grow 3D organoids. Carboplatin (50 μM), ReACp53 (4 μM), or the combination of the two agents was administered daily for 72 h. Treated organoids from some wells were harvested from Matrigel using 5 mg/mL dispase (Gibco, Waltham, MA, USA) for 30 min at 37 °C. The level of extracellular annexin V was assessed using a FITC Annexin V Apoptosis Detection Kit (BD Pharmagen, Franklin Lakes, NJ, USA). Staurosporine or DMSO-treated organoids were used as gating controls. Stained cells were analyzed using a BD LSRII flow cytometer and FACSDiva software (BD Biosciences, Franklin Lakes, NJ, USA). 

Other wells of similarly treated OVCAR3 organoids were harvested, lysed using RIPA buffer supplemented with protease inhibitor cocktail (ThermoFisher Scientific, Waltham, MA, USA), and used for Western blot analysis to measure the level of apoptosis biomarkers, cleaved PARP, and cleaved caspase 3. Total protein concentration of the lysate was measured using a BCA protein assay kit (ThermoFisher Scientific). Thirty micrograms of lysate per lane were loaded and run on 4–12% Bis-Tris polyacrylamide gels (Invitrogen, Waltham, MA, USA) and transferred to nitrocellulose membranes (Millipore, Burlington, MA, USA). Nonspecific antibody binding on membranes was blocked using phosphate-buffered saline containing 0.1% Tween and 5% non-fat dried milk (PBST+5% NFDM) and incubated for 1 h at room temperature. Blocked membranes were incubated with primary antibody diluted 1:1000 in PBST+5% NFDM overnight at 4 °C. Membranes were then washed in PBST and incubated in secondary antibody diluted 1:1000 in PBST+5% NFDM for 1 h at room temperature. Membranes were washed a final time using PBST and treated with Immobilon chemiluminescence substrate (Millipore, WBKLS0500). Protein was detected using a Biorad ChemiDoc Imaging System. Primary antibodies were the following: anti-PARP (Cell Signaling Technology, 9532, Topsfield, MA, USA); anti-Caspase 3 (Cell Signaling Technology, 9662), and anti-GAPDH (Cell Signaling Technology, 5174). The secondary antibody was HRP-linked anti-rabbit IgG (Cell Signaling Technology, 7074). The relative quantification of protein bands was done by densitometric analysis using ImageJ software. 

### 2.5. Animals

All animal experiments were approved by the UCLA Animal Research Committee (protocol 2008-153), the West Los Angeles Veterans Administration Medical Center IACUC (protocol 2019-020090) and performed under the oversight of the UCLA Division of Laboratory Animal Medicine. Female NSG mice (NOD.Cg-Prkdc^scid^Il2rg^tm1Wjl^/SzJ, Jackson Laboratory) ages 6–8 weeks were used for all in vivo experiments. All mice were housed in specific pathogen-free (SPF) facilities, in autoclaved cages with sterile bedding and food. 

### 2.6. Establishment of Intraperitoneal Xenografts

Intraperitoneal xenografts were established in female NSG mice using OVCAR3, SKOV3, and OVCAR8 ovarian cancer cell lines. Xenografts were established by injecting a 50/50 Matrigel/tumor cell suspension (100 μL total) into the IP space of mice using a 20-gauge needle. Two weeks after tumor establishment, mice were allocated into experimental groups by simple randomization, with one mouse randomly selected from each experimental cohort and euthanized to confirm tumor take. 

### 2.7. Quantification of Tumor Burden In Vivo

Intraperitoneal disease burden based on peritoneal lavage: Pelvic washes were performed using RPMI medium (Gibco) on mice following euthanasia. Cells obtained from pelvic washings were harvested and enzymatically digested with collagenase and dispase (1 mg/mL each, Gibco) for 45–60 min at 37 °C. Digested cell pellets were resuspended in RPMI and quantified either by manual counting on a hemocytometer by two independent investigators or using an automated cell counter (Countess II, ThermoFisher Scientific). Values reported reflect the total number of cells recovered from peritoneal washings.

Intraperitoneal disease burden based on tumor implant counts: The number of tumor implants was quantified by two independent investigators using histologic sections of harvested mouse organs immunohistochemically stained for either p53 or Pax8, at five levels throughout the depth of the tissue block (Appendix A). The number of tumor implants per level was summed to give the total number of tumor implants per mouse and then averaged across treatment groups. A tumor implant was defined as a continuous region of tumor cells that exhibited positive staining for p53 (for OVCAR3 tumors with aggregating p53 mutations) or Pax8 (for SKOV3 tumors known to be p53-null). Non-continuous regions of positive staining that were near one another were counted as separate tumor implants (Appendix A).

### 2.8. Flow Cytometry to Determine Percentage of Tumor Cells from Peritoneal Lavage

Immunostaining of mouse pelvic wash cells for Trop1 expression was performed by incubating cells in anti-CD326 PE-Cyanine7 conjugated antibody (Invitrogen) at a concentration of 5 μL per million cells for 30 min. Stained cells were analyzed using a BD FACSCelesta and FACSDiva software (BD Biosciences, Franklin Lakes, NJ, USA).

### 2.9. Immunohistochemistry to Detect Tumor Cells

The primary antibodies used to detect tumor cells were anti-p53 (sc-126, clone DO-1, 1:250; Santa Cruz Biotechnology, Dallas, TX, USA) and anti-Pax8 (MRQ-50, 1:500; Cell Marque). The secondary antibodies used were biotinylated rabbit anti-mouse (Jackson Immunoresearch, 1:1000), and the tertiary antibody used was streptavidin-conjugated horseradish peroxidase (Jackson Immunoresearch, 1:1000). Detection was performed using 3,3′-diaminobenzidine (DAB) chromagen (HK130-5K, Biogenex).

### 2.10. Determination of p53 Mutation Status in Patient Samples and Cell Lines

p53 mutation status for each patient sample was determined by clinical sequencing of the primary tumor (FoundationOne, Cambridge, MA, USA) or whole-exome sequencing. For whole-exome sequencing: the Kapa Hyper library kit was used to make the genomic DNA library. The workflow consisted of fragmentation of gDNA, end repair to generate blunt ends, A-tailing, adaptor ligation, and PCR amplification. Different indices were used for multiplexing samples in one lane. Whole-exome DNA was captured from total genomic DNA using the SeqCap EZ System from NimbleGen according to the manufacturer’s instructions. Briefly, the gDNA library was incubated with SeqCap biotinylated DNA baits, and the hybrids were purified using streptavidin-coated magnetic beads, then followed by PCR. Sequencing was performed on an Illumina HiSeq3000 for a pair-end 150 bp run. 

Raw reads were mapped to the GRch38 human genome reference assembly (GCA_000001405.15) using BWA-MEM. Variants were called using GATK v4.0.10.0 [21] according to GATK’s best practices for somatic mutation calling using the Mutect2 tool. Variant calling in cancer cells was aided by using a panel of normal variants (PON) generated from patient-matched PBMCs and the af-only-gnomad.hg38.vcf germline resources and then the called somatic variants were filtered according to GATK recommendations. TP53 mutation status for each cell line was obtained from the International Agency for Research on Cancer (IARC) p53 database (Version R20) [22]. 

### 2.11. Synergy Analysis

Synergy scores using four separate reference models (the Highest Single Agent (HSA) model, the Loewe Additivity model, the Bliss Independence model, and the Zero Interaction Potency (ZIP) model) [23] were calculated for each cell line and patient sample tested in the high throughput in vitro organoid drug assay using the computational tool SynergyFinder 2.0, which has an R-package as well as a web application [24]. The synergistic or antagonistic effect of the pairwise combination of doses was visualized as two-dimensional synergy/antagonism heatmaps and then summarized over the full dose–response matrix using a synergy score calculated for each reference model. Experimental results were run in triplicate, and results were averaged for analytical purposes. SynergyFinder 2.0 allows for inputting data from independent replicate experiments in order to calculate a 95% confidence interval for synergy scoring [24]. Based on these reference models, a summary synergy score value >10 is considered synergistic, between −10 and +10 is considered additive, and a synergy score <−10 is considered antagonistic [24]. We considered each sample as synergistic, additive, or antagonistic based upon the majority results from the four models available in SynergyFinder (e.g., if 3 or more models agreed, the combination was synergistic).

### 2.12. Statistics

Results are reported as means ± standard deviation. The *p*-values for comparing means were computed using one-way analysis of variance, allowing for non-constant variance (variance heterogeneity). The Shapiro–Wilkes test on the residual errors was computed to confirm normality. When data was not normal, *p*-values were computed with the non-parametric Kruskal–Wallis test. Calculations were carried out using R version 4.0.5 (R Foundation for Statistical Computing, Vienna, Austria). Survival curves were estimated with the Kaplan–Meier method and were compared between groups using the log-rank test. *p* < 0.05 were considered statistically significant. IC50 values were calculated using the four-parameter logistic model and GraphPad Prism 8.

## 3. Results

### 3.1. A Smaller Disease Burden Was Found in Ovarian Cancer Bearing Mice with Administration of ReACp53 Compared to Vehicle

ReACp53 has been shown to target subcuticular xenografts harboring aggregating p53 mutations in prostate and pancreatic tumor models when administered every 48 h [14,15]. In these same studies, xenografts bearing wildtype p53 were not targeted with this peptide [14,15]. HGSOC is a disease that primarily metastasizes in the peritoneal cavity. We, therefore, sought to test if ReACp53 could effectively target HGSOC tumor cells in vivo when administered at a frequency of 3×/week using an intraperitoneal (IP) model of ovarian cancer. An IP tumor model was selected for these studies as it closely mimics the disease spread observed in ovarian cancer patients, including the development of ascites and multiple metastatic implants on peritoneal organs. Given the half-life of peptides is generally short [25], daily administration of ReACp53 was compared to a more clinically feasible dosing regimen of 3×/week. OVCAR3 xenografts were established in n = 13 NSG mice (Figure 1). Two weeks following tumor establishment, n = 1 mouse was euthanized to confirm tumor take (Appendix A). The remaining n = 12 mice were randomized into one of three treatment groups and received either vehicle, ReACp53 15 mg/kg 7×/week, or ReACp53 15 mg/kg 3×/week (n = 4/treatment group, Figure 1A). Mice were euthanized after three weeks of treatment and residual tumor cells were harvested by peritoneal lavage (Figure 1B, Appendix A) and quantified (Figure 1C). Results demonstrated a reduction in total numbers of peritoneal cells in mice treated with ReACp53 (administered 7×/week or 3×/week) compared to vehicle (*p* < 0.05, Figure 1C). A flow cytometry analysis was performed to measure the percentage of Trop1+ cells in pelvic washings. These results demonstrated that the majority of cells retrieved from pelvic washings were epithelial, and likely tumor cells (Appendix A).

Organs from euthanized mice were harvested, formalin-fixed and paraffin-embedded (FFPE) for histologic analysis. FFPE organs were thoroughly sampled at 5 levels, 100 µM apart, throughout the depth of the tissue block. Slides from each level were stained for p53 to identify tumor implants (Appendix A). Implants were quantified by manual counting, summed across all five levels for each mouse, and averaged by treatment group (Figure 1D). The average number of tumor implants was significantly decreased in mice treated with ReACp53 when administered 7×/week or at a reduced frequency of 3×/week compared to vehicle (*p* < 0.05, Figure 1D). The same dose of ReACp53 (15 mg/kg 3×/week) administered for 4 weeks did not target p53-null SKOV3 IP xenografts (Appendix A). A similar approach for the quantification of tumor burden in SKOV3 tumor-bearing mice was utilized for this experiment (Appendix A).

Overall, these results demonstrated the administration of ReACp53 resulted in a reduced tumor burden in mice bearing OVCAR3 human ovarian cancer cells, but not p53-null SKOV3 cells using a physiologic intraperitoneal disease model.

### 3.2. Resurgence of Disease Was Observed after Cessation of ReACp53 Administration

We went on to test the long-term efficacy of ReACp53 in targeting OVCAR3 tumors using the same intraperitoneal model. Here, the disease burden was evaluated after a period of time following the cessation of ReACp53 administration (Figure 2). NSG mice were injected with 1.0 × 10^6^ OVCAR3 cells into the IP space of n = 17 animals (Figure 2A). Following two weeks of tumor establishment, one mouse was euthanized to confirm tumor take (Appendix A). The remaining n = 16 animals were randomized to receive either vehicle or ReACp53 15 mg/kg 3×/week (n = 8/treatment group) for four weeks. n = 3 mice/treatment group were euthanized soon after treatment (immediately post-therapy cohort). The remaining mice were kept off-therapy for four weeks and then euthanized to assess any resurgent disease (release cohort).

Data for the immediately post-therapy cohort is shown in Figure 2B–D. Mice treated with ReACp53 had less tumor burden compared to vehicle-treated mice (Figure 2B–D, Appendix A). In mice euthanized 4 weeks post-treatment (release cohort), a resurgence of disease was observed despite administration of ReACp53 as evidenced by equivalent numbers of intraperitoneal tumor cells (*p* = 0.06) and organ implants (*p* = 0.56) in ReACp53-treated compared to vehicle-treated mice. (Figure 2E–G, Appendix A).

### 3.3. ReACp53 and Carboplatin Exhibited Synergistic Activity in Targeting a Subset of Human Ovarian Cancer Cell Lines In Vitro

Our results demonstrated that ovarian cancer xenografts relapse after the cessation of ReACp53 administration (Figure 2). Carboplatin is a cytotoxic DNA-damaging drug that is used as part of the standard chemotherapy for patients diagnosed with ovarian cancers, but in many cases, tumors relapse despite this treatment [3]. We, therefore, sought to test if there was any potential synergistic activity in targeting ovarian cancer cells when combining ReACp53 with carboplatin. 

We first tested this combinatorial approach using a panel of commercially available ovarian cancer cell lines with varying levels of platinum sensitivity [26,27], as outlined in Appendix A. To assess any potential synergy in the ReACp53 and carboplatin combination, a high throughput in vitro organoid drug assay [19] was utilized to determine drug sensitivity at various dose combinations (Figure 3A). In this assay, cell viability was assessed using an ATP luminescent substrate [19]. Cell viability values were utilized to calculate drug synergy in four separate reference models, including the Highest Single Agent (HSA) model, the Loewe additivity model, the Bliss Independence model, and the Zero Interaction Potency (ZIP) model, calculated using the SynergyFinder 2.0 tool [24]. Two out of eight CCLE ovarian cancer cell lines tested in this assay demonstrated synergy (Figure 3B). The remaining six cell lines demonstrated an additive effect for the ReACp53 and carboplatin combination (Figure 3B). TP53 mutation status for each cell line was obtained from the International Agency for Research on Cancer (IARC) p53 database [22] and reported in Figure 3B. Complete viability plots and for each cell line are reported in Appendix A.

To explore potential mechanisms of synergy observed in cells lines treated with carboplatin and ReACp53, OVCAR3 organoids were treated with vehicle, ReACp53 (4 μM), carboplatin (50 μM), or the combination of the two agents for 72 h. The drugs were replenished daily. Treated organoids were harvested, stained with annexin V and propidium iodide, and analyzed by flow cytometry. Increased apoptosis measured by annexin V was seen in all treatment groups compared to vehicle, with the highest level detected in organoids treated with the combination (Figure 3C,D). Western blot analysis demonstrated a trend toward increased cleaved PARP and cleaved caspase 3 protein in organoids treated with the combination (Figure 3E, Appendix A). Collectively, results suggest there is increased apoptosis when ReACp53 is combined with carboplatin in targeting OVCAR3 cells.

### 3.4. Impact of ReACp53 and Carboplatin on Survival Using an In Vivo IP Model of Human Ovarian Cancer

To further assess the potential tumor targeting of the ReACp53 and carboplatin combination, the in vivo efficacy of these two agents was tested in mice bearing ovarian cancer cell line xenografts in a survival analysis. We selected two cell lines for this study based on results from the in vitro organoid drug assay. OVCAR3 cells demonstrated synergy when ReACp53 was added to carboplatin, whereas OVACR8 cells demonstrated additive effects for the combination (Figure 3B). 

IP xenografts were established in NSG mice using OVCAR3 cells in n = 29 animals. Following two weeks of tumor establishment, one mouse was euthanized to confirm tumor take (Appendix A). The remaining n = 28 mice were randomized to receive a four-week course of either vehicle, carboplatin, ReACp53, or ReACp53 and carboplatin (n = 7/treatment group). In all survival experiments, the sequence of drug administration for animals treated with the combination was ReACp53 followed by administration of carboplatin an hour later. The rationale for this sequence was to optimize ReACp53-mediated mitochondrial cell death as reported by others [14], prior to carboplatin-induced DNA damage. OVCAR3 cells are known to be platinum-sensitive [26]; therefore, in these experiments, tumor-bearing mice were treated with carboplatin at a dose of 10 mg/kg IP 1×/week. ReACp53 was injected at a dose of 15 mg/kg IP 3×/week. After four weeks of treatment, mice were released off-therapy and euthanized only upon reaching NIH-defined endpoint criteria [28]. Large ascites burden, hunched posture, reduction in locomotion, and matted and unkempt appearance were the primary reasons for euthanasia in this cohort of animals. The time from injection of tumor cells to endpoint was recorded for each mouse, and overall survival was calculated and compared between treatment groups (Figure 4A). The median overall survival for mice treated with ReACp53 and carboplatin combination (157 days) was extended compared to ReACp53 (95 days, *p* < 0.001), carboplatin (131 days, *p* < 0.001), or vehicle (97 days, *p* < 0.001) (Figure 4A). Stable mouse weights recorded throughout the treatment phase indicate the tolerability of ReACp53 and carboplatin combination treatment (Appendix A). 

To further test the efficacy of this ReACp53 and carboplatin, we next established xenografts using OVCAR8 cells in n = 29 NSG mice. Tumor establishment was confirmed in n = 1 mouse euthanized two weeks after OVCAR8 cell injection (Appendix A). Given that OVCAR8 cells are platinum-resistant [27], a dose of 50 mg/kg carboplatin was used in this experiment. Twenty-eight OVCAR8-bearing mice were randomized to receive treatment with either vehicle, carboplatin, ReACp53, or ReACp53 and carboplatin for four weeks (n = 7 mice/treatment). Similarly, mice were euthanized upon reaching NIH-defined endpoint criteria [28]. Mice treated with the ReACp53 and carboplatin combination had an increase in median survival (52 days) compared to vehicle (46 days, *p* = 0.024) and ReACp53 (46 days, *p* = 0.006). However, the addition of ReACp53 to carboplatin did not extend survival compared to carboplatin treatment alone (47 days, *p* = 0.456) (Figure 4B). In this cohort, the vast majority of mice reached the endpoint due to severe and progressive weight loss (Appendix A). Due to the aggressive nature of OVCAR8 tumor cells, some mice did not complete the full four-week treatment.

### 3.5. Addition of ReACp53 to Carboplatin May Enhance Tumor Cell Targeting of Primary Patient HGSOCs

To further explore whether the addition of ReACp53 to carboplatin can enhance tumor cell targeting of human ovarian cancers, ten independent primary HGSOC tumor samples were tested using the high throughput in vitro organoid drug assay. These samples comprised chemo naïve (n = 4), chemotherapy-treated (n = 4), and recurrent tumors (n = 2) (Figure 5A). For these specimens, clinical data was used to determine the sensitivity of patients’ tumors to platinum drugs (Figure 5A). Based on analysis using the SynergyFinder tool, the combination of ReACp53 and carboplatin demonstrated additive effects in 8/10 patient tumors tested and antagonistic effects in 1/10 patient samples (Figure 5B). For one sample (HGSOC2), two models predicted an additive effect for the ReACp53 and carboplatin combination, while two models predicted antagonistic effects of these two drugs (Figure 5B). A complete dataset of viability plots for each patient tumor sample is reported in Appendix A.

## 4. Discussion

Carboplatin is the frontline treatment for patients diagnosed with HGSOC [3]. While the initial response to this chemotherapy is favorable, the majority of patients experience tumor recurrence associated with the development of platinum-resistant disease. The underlying biologic cause for platinum resistance has been explored for decades and is likely a multifactorial process mediated by drug transport, tolerance of cancer cells to DNA damage, loss of p53 function, and evasion of apoptosis, among others [29]. Mutations in p53 are also associated with resistance to chemotherapy in many cancers and worse potential outcomes [12,13]. In fact, alterations in p53 are frequently found in castration-resistant prostate cancer (CRPC) and may be enriched in advanced, metastatic disease compared to primary prostate cancer [30]. 

The efficacy of ReACp53 in targeting tumor cells has been evaluated by other researchers. For example, ReACp53 was effective in targeting prostate cancer cells with aggregating mutations in p53 by increasing mitochondrial cell death and inhibiting DNA synthesis [14]. This peptide also inhibited xenograft growth of prostate cancer tumors in vivo [14]. ReACp53 could also prohibit the growth of pancreatic cancer xenografts harboring aggregating mutations in p53 [15]. Recent work suggests that ReACp53 can also restore sensitivity to cisplatin in a human lung cancer cell line expressing an exogenous p53R282W mutation in vitro [31]. 

Our study focused on the analysis of epithelial ovarian cancer samples known to frequently carry mutations in p53 [6]. Given the prevalence of p53 mutations observed in HGSOCs, these tumors are an ideal model for further testing ReACp53 peptide alone and in combination with standard carboplatin chemotherapy. Synergy analysis was used to determine relative sensitivity to the ReACp53 and carboplatin combination. Notably, for 17/18 (94%) total samples tested, including both cell lines and patient tumor samples, results from at least three out of four synergy models agreed. In the majority of primary HGSOC tumor samples tested, an additive effect for the combination was observed, indicating that tumor targeting may be enhanced when adding ReACp53 to carboplatin. These samples included patients with clinically-determined platinum-sensitive and -resistant disease. Importantly, in 5/6 samples from patients with platinum-resistant disease, the addition of ReACp53 enhanced the efficacy of carboplatin when administered in combination, suggesting this approach may be applicable in targeting therapy-resistant tumors.

In the four platinum-resistant ovarian cancer cell lines tested, the combination of ReACp53 and carboplatin yielded additive effects. In OVCAR4 wells with intermediate sensitivity to platinum drugs, synergy was observed. Within the tested cell lines, the highest level of synergy was seen when ReACp53 and carboplatin were combined in targeting OVCAR3 cells. In these cells, reduced cell viability may be mediated through enhanced apoptosis. Potential synergy in carboplatin and ReACp53 was validated using OVCAR3 cells in an in vivo survival analysis. Given that an additive effect was seen when ReACp53 was added to carboplatin in OVCAR8 cells, the in vivo efficacy of this combination was tested using this ovarian cancer cell line as well. Here, an improvement in survival was not observed when the combination was administered compared to carboplatin treatment alone. A challenge with this in vivo survival study was the aggressive nature of OVCAR8 cells that required euthanizing a subset of the mice prior to, or soon after, administration of ReACp53 and carboplatin. Some mice in this cohort could not complete the four-week course of treatment; hence, OVCAR8 cells may not be an optimal model for survival analysis. Collectively, our data suggest that ReACp53 in combination with carboplatin may provide a potential strategy for targeting human epithelial ovarian cancers. Given the tolerability of ReACp53 and carboplatin observed in vivo, further analyses into the potential of this combinatorial strategy may be warranted.

While ReACp53 is designed to specifically target mutant aggregating p53 protein, there is emerging data that it may have cytotoxic effects through other mechanisms, including targeting p53 interactions with p63, p73, or the wildtype protein [17,32]. There is some evidence that in malignant hematopoietic cell lines, ReACp53 can target cell lines both with and without aggregating p53 mutations in a p73-dependent manner [32]. Additionally, it is also suggested that ReACp53 may impact cell cycle transition in malignant cells [14]. Work from others suggests that ReACp53 and other peptides designed to target p53 aggregation may have p53-dependent and independent effects [17]. In this study, we observed that ReACp53 could potentially target cell lines and tumors with p53 mutations not known to result in aggregation (Appendix A), suggesting that other potential mechanisms of ReACp53 action may be present in targeting tumors as reported by other investigators [17,32]. 

ReACp53 is a peptide. Therapeutic peptides have some advantages, including potentially less toxicity, increased specificity, and reduced side effects, compared to other small molecules and cytotoxic drugs [25]. However, the effective use of peptides for cancer therapy is hindered by many challenges, including a metabolic breakdown and fast clearance in vivo [25]. Such inherent challenges in the administration of peptide-based therapeutics must be addressed in order to further test the viability of ReACp53 for clinical use. Thus far, we have tested a frequent intraperitoneal route of administration for ReACp53, a strategy that is feasible but not performed frequently in patients. Other well-tolerated p53 targeting peptides administered intraperitoneally, such as ADH-6, have similarly demonstrated efficacy in targeting p53-mutated cancers [15]. In future work, consideration can be given to other routes of delivery, including intravenous infusion or strategies being tested for nanoparticle delivery of peptides [33]. 

Overall, the findings presented here demonstrated that the addition of ReACp53 to carboplatin may enhance targeting in a subset of HGSOC tumors. The ReACp53 and carboplatin combination demonstrated synergistic effects in some ovarian cancer cell lines and additive effects in primary HGSOC tumor samples tested.

## 5. Conclusions

Clinical development of targeted approaches and novel combination therapies for the treatment of ovarian cancer remains a major goal of researchers globally. Results from this study indicated that the addition of ReACp53 to standard carboplatin chemotherapy can enhance tumor targeting in a subset of ovarian tumors.

## Figures and Tables

**Figure 1 cancers-13-05908-f001:**
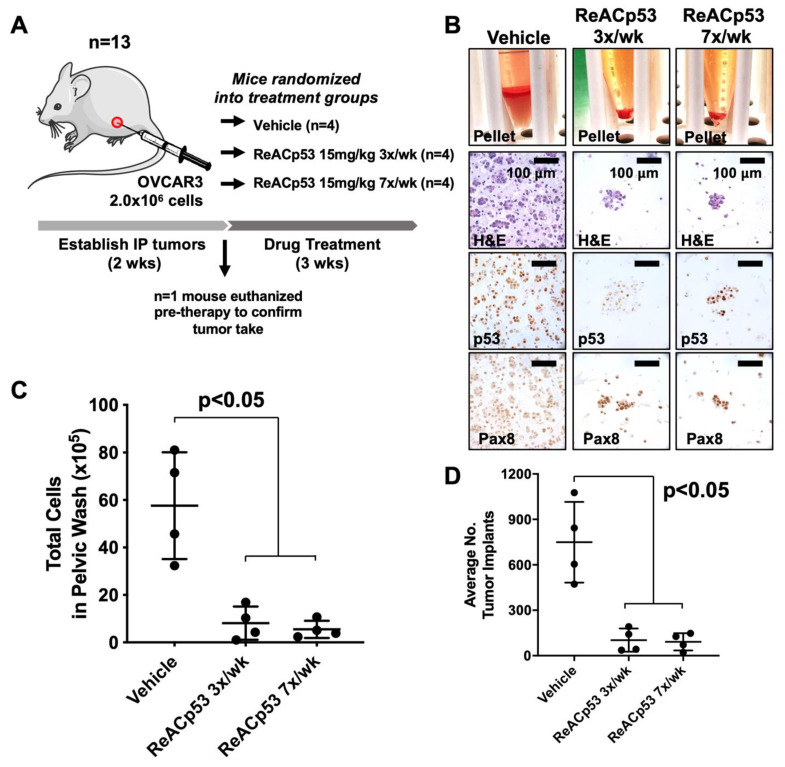
Ovarian cancer xenograft-bearing mice injected with ReACp53 had a smaller disease burden compared to vehicle. (**A**) Xenografts were established by injecting 2.0 × 10^6^ OVCAR3 cells into the intraperitoneal (IP) space of n = 13 NSG mice. Following two weeks of tumor establishment, n = 1 mouse was euthanized to confirm tumor take. The remaining n = 12 mice were randomized to receive either vehicle or ReACp53 15 mg/kg (administered 3×/week or 7×/week, IP) for three weeks. (**B**) At the end of therapy, mice were euthanized, IP tumors were harvested by peritoneal lavage, and harvested cells were immunostained for p53 and Pax8 to confirm the presence of tumor cells. Representative cell pellets are shown. (**C**) The total number of cells harvested was quantified for each mouse. Results demonstrated a significant reduction in the number of cells in mice treated with ReACp53 (either 3×/week or 7×/week) compared to vehicle treatment (*p* < 0.05). (**D**) Organs harvested from euthanized mice were histologically examined, and the number of tumor implants was quantified across five independent levels per animal and averaged by treatment group (n = 4 animals/group). The average number of tumor implants was significantly reduced in mice treated with ReACp53 (either 3×/week or 7×/week) compared to vehicle (*p* < 0.05).

**Figure 2 cancers-13-05908-f002:**
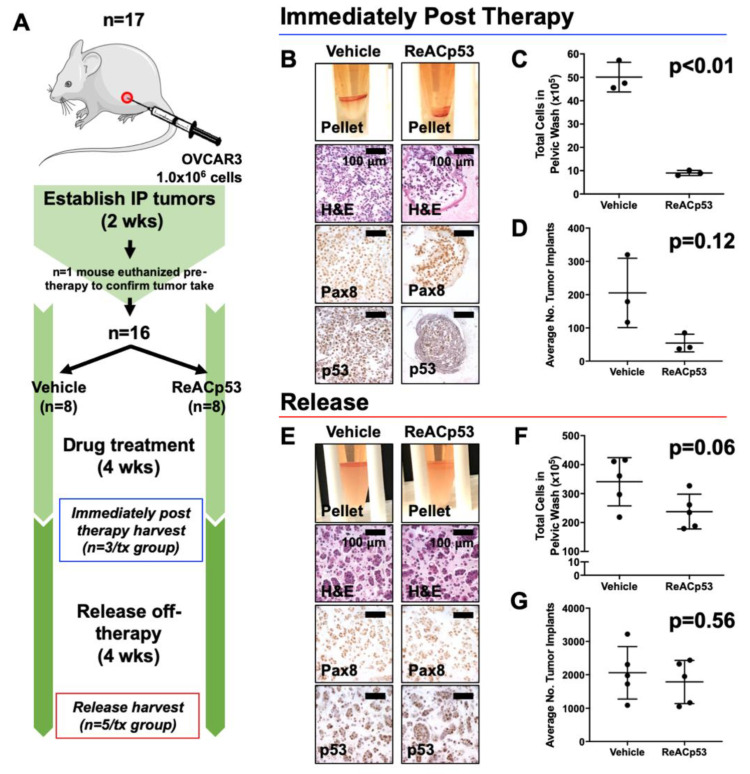
Resurgence of tumors after cessation of ReACp53 therapy in vivo. (**A**) Xenografts were established by injecting 1.0 × 10^6^ OVCAR3 cells into the intraperitoneal (IP) space of n = 17 NSG mice. Tumor take was confirmed in n = 1 mouse after two weeks of tumor establishment. The remaining mice (n = 16) were randomized to receive either vehicle or ReACp53 15 mg/kg 3×/week IP for four weeks (n = 8/treatment). A cohort of mice was harvested after four weeks of treatment (n = 3/treatment, immediately post-therapy cohort). The remaining mice were released off-therapy and euthanized four weeks later (n = 5/treatment, release cohort). (**B**–**D**) Results from mice harvested immediately post-therapy. (**B**) Representative cell pellets harvested from euthanized mice. Tumors cells were confirmed by Pax8 and p53 staining. **(C)** The total number of cells harvested was quantified for each mouse. Results demonstrated a lower tumor burden in mice treated with ReACp53 vs. vehicle (*p* < 0.01). (**D**) The average number of tumor implants was lower in mice treated with ReACp53 vs. vehicle, though results did not reach statistical significance (*p* = 0.12). (**E**–**G**) Results from mice harvested after 4 weeks release off-therapy. (**E**) Representative images of cell pellets. Immunostaining for Pax8 and p53 confirmed the presence of tumor cells. (**F**) Quantification of harvested IP cells demonstrated the resurgence of tumors in mice treated with ReACp53 (*p* = 0.06). (**G**) The average number of tumor implants was equivalent in mice treated with ReACp53 vs. vehicle (*p* = 0.56).

**Figure 3 cancers-13-05908-f003:**
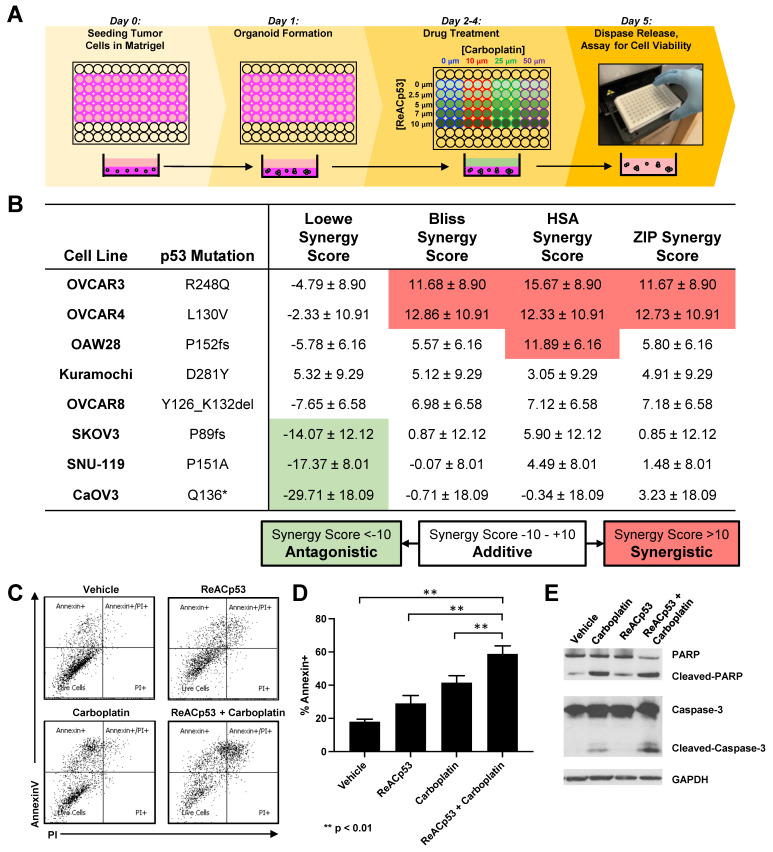
Synergistic activity of ReACp53 and carboplatin combination observed in a subset of human ovarian cancer cell lines in vitro. (**A**) Schema of the in vitro 3D mini-ring organoid drug assay. Drug interaction studies were performed across a range of ReACp53 (0–10 µM) and carboplatin (0–50 µM) concentrations. (**B**) Eight independent ovarian cancer cell lines annotated in the Cancer Cell Line Encyclopedia (CCLE) were tested using the 3D mini-ring organoid drug assay, and potential synergy for the ReACp53 and carboplatin combination was calculated. Data shown were calculated using SynergyFinder 2.0 to measure synergy score ± 95% confidence interval. Results were averaged from five independent experiments plated by two separate investigators. In this analysis, OVCAR3 and OVCAR4 cells exhibited synergy when treated with the ReACp53 and carboplatin combination for the majority of the synergy models assessed (Loewe, Bliss, HSA, and ZIP). The remaining cell lines (OAW28, Kuramochi, OVCAR8, SKOV3, SNU-119, and CaOV3) exhibited additive effects for ReACp53 and carboplatin combination for the majority of synergy models assessed. (**C**) OVCAR3 organoids were treated with vehicle, ReACp53 (4 μM), carboplatin (50 μM), or ReACp53 + carboplatin for 72 h with daily drug replenishment. Organoids were released from Matrigel and stained for annexin V and propidium iodide. Data from one experiment is shown. (**D**) Percentage of annexin V+ cells in each treatment group. Data represent the mean of three independent experiments. ** *p* < 0.01. (**E**) Representative Western blot for detection of PARP, caspase 3, and GAPDH loading control.

**Figure 4 cancers-13-05908-f004:**
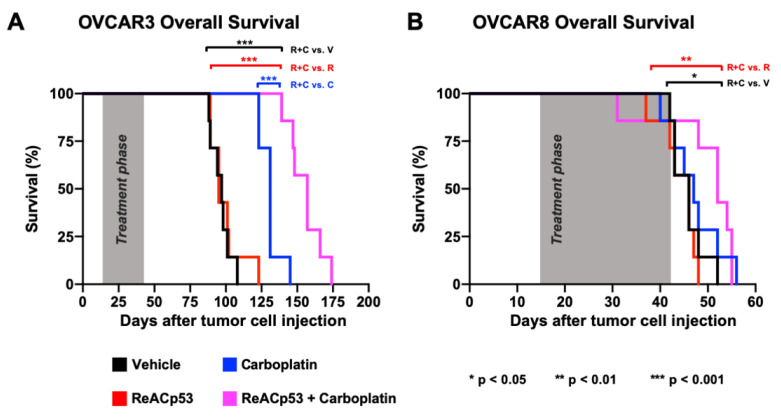
ReACp53 to carboplatin combination extended overall survival of mice bearing OVCAR3 but not OVCAR8 ovarian cancer xenografts. Xenografts were established by injecting either 3.0 × 10^6^ OVCAR3 or OVCAR8 cells into the intraperitoneal (IP) space of NSG mice (n = 29 mice/cell line). We confirmed tumor take by euthanizing n = 1 mouse/cell line prior to initiating treatment. The remaining n = 28 mice/cell line were randomized to receive either vehicle, ReACp53 15 mg/kg 3×/week IP, carboplatin (10 mg/kg for OVCAR3 tumors, 50 mg/kg for OVCAR8 tumors) 1×/week IP, or ReACp53 and carboplatin combination therapy (n = 7/treatment). Following four weeks of treatment, mice were released off-therapy and monitored daily for signs of distress. Upon reaching NIH-endpoint criteria, mice were euthanized, and the total time from tumor cell injection to endpoint was recorded for each animal. These data were used to generate Kaplan–Meier curves for overall survival. (**A**) OVCAR3 tumor-bearing mice treated with ReACp53 and carboplatin combination therapy had a longer median survival (157 days) compared to vehicle (97 days, *p* < 0.001), ReACp53 (95 days, *p* < 0.001), or carboplatin (131 days, *p* < 0.001). (**B**) OVCAR8 tumor-bearing mice treated with ReACp53 and carboplatin combination therapy had a median survival of 52 days compared to vehicle (46 days, *p* < 0.05), ReACp53 (46 days, *p* < 0.01), or carboplatin (47 days, *p* = 0.46). * *p* < 0.05, ** *p* < 0.01, *** *p* < 0.001.

**Figure 5 cancers-13-05908-f005:**
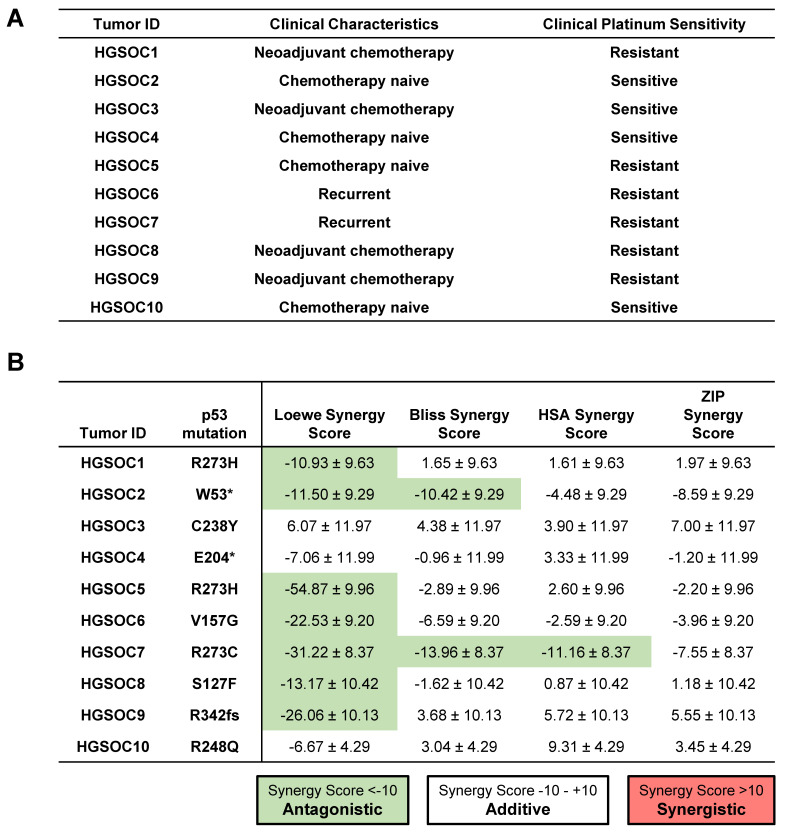
Addition of ReACp53 to carboplatin may enhance tumor cell targeting of primary patient HGSOCs. (**A**) Clinical characteristics and platinum sensitivity for each patient sample are shown. (**B**) Cryopreserved dissociated high-grade serous ovarian tumors (or ascites) were plated in the in vitro 3D mini-ring organoid drug assay and treated with various doses of ReACp53 and carboplatin. Cell viability data were used to construct synergy response surfaces and summarized as a synergy score using four separate synergy models (Loewe, Bliss, HSA, and ZIP). Data represent the synergy score ± 95% confidence interval, as calculated by SynergyFinder 2.0 based on the average of two independent experiments plated by separate investigators. Among the 10 HGSOC tumors tested with ReACp53 and carboplatin, eight demonstrated additive effects, one exhibited antagonism, and one (HGSOC2) was undetermined based on the results of the four synergy models assessed. Mutation status in p53 was verified using whole-exome sequencing or clinical sequencing.

## Data Availability

Data sharing is not applicable to this article.

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
