# Peer review of "Combining ReACp53 with Carboplatin to Target High-Grade Serous Ovarian Cancers"

_cancers, 2021, doi:10.3390/cancers13235908_

Round 1
Reviewer 1 Report
- Authors are advised to mention the mechanism of action of ReACp53 in introduction section.
- The Rationale for using 3X or 7X treatment of ReACp53 is not clear in Figure 1.
- Authors are advised to mention the dose of ReACp53 in figure 1.
- Authors are advised to measure the band intensity of different proteins in the western blot image in Fig 3E
Author Response
Response to reviewer’s comments
Thank you for your thorough review. Your suggestions have improved the quality of our manuscript.
- Authors are advised to mention the mechanism of action of ReACp53 in introduction section.
- Thank you for pointing this out. The introduction has been revised to address this question. Please see line 90-92
- The Rationale for using 3X or 7X treatment of ReACp53 is not clear in Figure 1.
- Thank you for pointing this out. Our rationale has been added to section 3.1 of the manuscript. Please see lines 269-271
- Authors are advised to mention the dose of ReACp53 in figure 1.
- Thank you for pointing this out. Revisions are made to figure 1 outlining the dose.
- Authors are advised to measure the band intensity of different proteins in the western blot image in Fig 3E.
- Thank you for pointing this out. This data is incorporated in supplementary figure 7. We have modified the figure and the legend and clarified that run 1 of experiments is shown in figure 3E. Run 2 and run 3 are shown in supplementary figure 7. Quantification based on band intensity averaged from the 3 separate experiments is shown in supplementary figure 7C. Additional clarification is made in method section 2.04. Please see line 164-166.

Reviewer 2 Report
The manuscript by Neal et al. entitled “Combining ReACp53 with carboplatin in targeting high grade serous ovarian cancers” examines the use of standard platinum-based chemotherapy with a peptide designed to inhibit mutant p53 aggregation (ReACp53) to target human ovarian tumors. Overall, the study is well-written and provides important information on the use of combinatorial approaches in cancer therapy, specifically in regards to restoration of p53 function. While the authors state that ReACp53 has been shown to target multiple cancer models in vitro and in vivo, the current study focuses on its effectiveness in targeting human ovarian cancers. A few minor concerns are described below.
- As figure 3 shows very little (if any) apoptosis occurring in the presence of ReACp53 alone, and when considered in the context of the reduced disease burden in the presence of ReACp53 as shown in figure 1, have the authors consider the ability of ReACp53 to result in a cell cycle block in these cells?
- The viability data for the human HGSOC tumors currently shown in supplementary figure 10 provides important insight into the effectiveness of the combinational therapy. The authors should consider presenting this data in some form in the main body of the manuscript. Overwhelming in its current form, a give dose (5 uM) of ReACp53 at a given level of carboplatin (50 uM) for each sample would be informative.
- What is a bit unclear and deserves further comment is what is meant by “antagonistic effects” in regards to tumor ID HGSOC7 as shown in figure 5.
Author Response
Response to reviewer’s comments
Thank you for your thorough review. Your suggestions have improved the quality of our manuscript.
Comment: The manuscript by Neal et al. entitled “Combining ReACp53 with carboplatin in targeting high grade serous ovarian cancers” examines the use of standard platinum-based chemotherapy with a peptide designed to inhibit mutant p53 aggregation (ReACp53) to target human ovarian tumors. Overall, the study is well-written and provides important information on the use of combinatorial approaches in cancer therapy, specifically in regards to restoration of p53 function. While the authors state that ReACp53 has been shown to target multiple cancer models in vitro and in vivo, the current study focuses on its effectiveness in targeting human ovarian cancers. A few minor concerns are described below.
1. As figure 3 shows very little (if any) apoptosis occurring in the presence of ReACp53 alone, and when considered in the context of the reduced disease burden in the presence of ReACp53 as shown in figure 1, have the authors consider the ability of ReACp53 to result in a cell cycle block in these cells?
- This is an excellent point. The exact mechanism of cell death induced by ReACp53 remains unknown. Please see additions to introduction (line numbers 90-92) and discussion (line numbers 543-545) where we discuss impact on cell cycle.
2. The viability data for the human HGSOC tumors currently shown in supplementary figure 10 provides important insight into the effectiveness of the combinational therapy. The authors should consider presenting this data in some form in the main body of the manuscript. Overwhelming in its current form, a give dose (5 uM) of ReACp53 at a given level of carboplatin (50 uM) for each sample would be informative.
- We agree with the reviewer as this is an important point. Given that we tested a large number of concentrations for a panel of 10 patient samples, we have presented the synergy scores in the main figure which essentially summarized drug interactions across all doses tested. The viability plots with multiple concentrations are in supplementary figure 10. Presenting a single dose combination may artificially imply combinatorial efficacy when it may not be present. That is why synergy scoring across a large combination of concentrations tested may be more representative of drug interactions.
3. What is a bit unclear and deserves further comment is what is meant by “antagonistic effects” in regards to tumor ID HGSOC7 as shown in figure 5.
- Thank you for this important question. In methods section 2.11 we have defined parameters for antagonism. Please see lines 242-247. We have considered each sample as synergistic, additive, or antagonistic based on majority results from all four models in SynergyFinder. For HGSOC7 three out of four models suggested antagonism, therefore the effects of the two drugs in this specific tumor are considered to be antagonistic.
